# Application of Quantitative-PCR to Monitor Netpen Sites in British Columbia (Canada) for *Tenacibaculum* Species

**DOI:** 10.3390/pathogens10040414

**Published:** 2021-04-01

**Authors:** Joseph P. Nowlan, Scott R. Britney, John S. Lumsden, Spencer Russell

**Affiliations:** 1Center of Innovation for Fish Health, Vancouver Island University, Nanaimo, BC V9R 5S5, Canada; Scott.Britney@viu.ca (S.R.B.); Spencer.Russell@viu.ca (S.R.); 2Department of Pathobiology, University of Guelph, Guelph, ON N1G 2W1, Canada; jsl@uoguelph.ca

**Keywords:** *Tenacibaculum*, *Tenacibaculum maritimum*, *Tenacibaculum dicentrarchi*, tenacibaculosis, mouthrot, qPCR, Atlantic salmon, netpen

## Abstract

*Tenacibaculum* are frequently detected from fish with tenacibaculosis at aquaculture sites; however, information on the ecology of these bacteria is sparse. Quantitative-PCR assays were used to detect *T. maritimum* and *T. dicentrarchi* at commercial Atlantic salmon (*Salmo salar*) netpen sites throughout several tenacibaculosis outbreaks. *T. dicentrarchi* and *T. maritimum* were identified in live fish, dead fish, other organisms associated with netpens, water samples and on inanimate substrates, which indicates a ubiquitous distribution around stocked netpen sites. Before an outbreak, *T. dicentrarchi* was found throughout the environment and from fish, and *T. maritimum* was infrequently identified. During an outbreak, increases in the bacterial load in were recorded and no differences were recorded after an outbreak supporting the observed recrudescence of mouthrot. More bacteria were recorded in the summer months, with more mortality events and antibiotic treatments, indicating that seasonality may influence tenacibaculosis; however, outbreaks occurred in both seasons. Relationships were identified between fish mortalities and antimicrobial use to water quality parameters (temperature, salinity, dissolved oxygen) (*p* < 0.05), but with low R^2^ values (<0.25), other variables are also involved. Furthermore, *Tenacibaculum* species appear to have a ubiquitous spatial and temporal distribution around stocked netpen sites, and with the potential to induce disease in Atlantic salmon, continued research is needed.

## 1. Introduction

Within Canada, from 2016 to 2018, approximately a billion dollars is generated annually from salmonid aquaculture [1]. A major salmon health issue in Canada has been attributed to bacteria from the genus *Tenacibaculum* [2]. These bacteria are typically yellow-pigmented, Gram-negative, filamentous, and several species are putative pathogens of tenacibaculosis in finfish and some bivalves [2]. Tenacibaculosis is typically characterized by changes in fish behaviour, yellow plaques or ulcers on epidermal surfaces and increased mortality [2,3,4,5,6].

Knowledge of *Tenacibaculum* ecology (e.g., impacts, hosts and distributions) is sparse and has been garnered through understanding disease caused by *Tenacibaculum* species; finfish and bivalve mortality events are most commonly described [2,3,7,8,9,10,11,12,13]. The bacteria are also found on numerous other metazoans [2], including crustaceans [14,15,16], cnidarians [17,18,19] and on marine mammals (i.e., Orca [*Orcinus orca*]) [20], all of which may vector or facilitate tenacibaculosis. *Tenacibaculum* sp. have also been recovered from marine waters [21,22,23] and biofilms [24,25]. The presence of *Tenacibaculum* sp. in biofilm and planktonic states likely has important implications for disease, as suggested for the related pathogenic freshwater bacteria, *F. psychrophilum* [26]. The global distribution of *Tenacibaculum* species is largely unknown, but several species are suspected to be cosmopolitan [2]. The location of *Tenacibaculum* bacteria around aquaculture netpen sites are also widely unknown. An understanding of the spatial and temporal distributions of *Tenacibaculum* species may aid in the management of tenacibaculosis. Complete genome sequencing [27,28] and multi-locus sequence analysis [29,30] have been used to identify *Tenacibaculum* species and strains, and both could be used to help understand bacterial distributions; however, both techniques are expensive and time-consuming. In contrast, quantitative-PCR (qPCR) is economical, quantitative and could be used to identify a baseline that may indicate when an outbreak is about to occur [31,32]. The present study describes the local distribution of *T. maritimum* and *T. dicentrarchi* at two commercial netpen sites in British Columbia (BC; Canada) using two separate quantitative-PCR (qPCR) assays before, during and after tenacibaculosis outbreaks.

## 2. Results

### 2.1. Bacterial Isolate Collection

Sixty-three isolates were collected from the Midsummer (MS) site (36 isolates from fish tissues and 27 isolates from water samples). Thirty-eight isolates were collected from the Larson Island (LI) site (21 isolates from fish tissues and 17 isolates from water samples and infrastructure swabs). Isolates were elongated bacilli to filamentous, yellow or cream-coloured and Gram-negative (Figure 1). No isolates were identified as *T. maritimum* (MAR assay [18]), while 25 isolates were identified as *T. dicentrarchi* (DICEN assay [31,32]); seven of the 25 isolates were sent for 16S rDNA sequencing and were confirmed to be *T. dicentrarchi* or *T. finnmarkense* (Table 1).

### 2.2. qPCR Results

#### 2.2.1. Midsummer and Larson Island

Of the 1166 samples (776 for MS, 387 for LI) and ~2332 quantitative-PCR (qPCR) tests, 308 positive qPCR results were recorded (244 from MS, 64 from LI) (Table 2). The log-number of bacteria (LNOB) detected using both the MAR [18] and DICEN [31,32] assays together was greater at the MS site compared to LI (*p* = 6.8 × 10^−3^, Table 3). Within each site, the log-number of *T. dicentrarchi* was greater compared to *T. maritimum* (*p* < 2.2 × 10^−16^, Figure 2; Table 3). Similar numbers of *T. dicentrarchi* were recorded between sites (*p* < 2.2 × 10^−16^, Figure 2, Table 3), while fewer *T. maritimum* were recorded at LI (*p* < 2.2 × 10^−16^, Figure 2, Table 3). For both assays and sites together, approximately a log-unit increase was detected during an outbreak (*p* = 1.2 × 10^−5^, Table 3). A log-unit increase was found in fish samples in comparison to environmental samples (*p* = 4.9 × 10^−8^, Table 3), where fish tissues and other organisms (OO) associated with netpens had greater numbers of bacteria compared to water samples and infrastructure swabs (*p* = 7.8 × 10^−13^, Table 3). Further statistical comparisons were performed separately for each assay and netpen site (Table 3). 

#### 2.2.2. Midsummer

There were less log-units of *T. dicentrarchi* detected before an outbreak compared with during an outbreak (*p* = 8.2 × 10^−3^, Table 3, Figure 3), but there were no differences for *T. maritimum* (*p* = 8.2 × 10^−1^, Table 3, Figure 3). Notably, however, neither *T. dicentrarchi* nor *T. maritimum* were detected before fish were introduced to the site. More *T. maritimum* and *T. dicentrarchi* were recorded in fish and from OO compared to water and infrastructure swabs (*p* ≤ 2.1 × 10^−3^, Table 3). The number of *T. dicentrarchi* from live fish and infrastructure swabs was not different between outbreak categories, but trends for increased numbers during an infection were recorded (*p* ≥ 9.7 × 10^−2^, Table 3, Figure 4). Dead fish, other organisms and water samples generated significant differences in *T. dicentrarchi* between outbreak categories (*p* ≤ 4.7 × 10^−2^, Table 3, Figure 4). *T. maritimum* in three sample types (water, infrastructure and OO) could not be compared using outbreak categories, as there were too few positive samples for statistical comparisons. There were no differences in the amount of *T. maritimum* based on the outbreak category in dead fish and live fish (*p* ≥ 9.1 × 10^−1^, Table 3, Figure 4). 

#### 2.2.3. Larson

Differences in the number of *T. dicentrarchi* between outbreak categories could not be interpreted with only one sample for the pre- and post-outbreak categories (Table 3). A 1.3 log increase in *T. maritimum* was identified during an outbreak (*p* = 6.4 × 10^−4^, Table 3). Like the MS site, more *T. dicentrarchi* and *T. maritimum* were identified in fish and OO than in water samples and infrastructure swabs (*p* ≤ 6.7 × 10^−4^, Table 3). In both assays, comparing the number of bacteria per sample type based on the outbreak category was not possible, as too few samples were positive. 

### 2.3. Supplemental Data Comparisons

At the MS site, there was no difference in the number of mortalities when *T. maritimum* (*p* = 7.5 × 10^−2^) or *T. dicentrarchi* (*p* = 3.2 × 10^−1^) were present. At the MS site, there were also more fish mortalities during netpen cleaning (*p* = 1.6 × 10^−4^) and when florfenicol was applied (*p* = 9.8 × 10^−7^). At the LI site, there was no difference comparing mortalities when *T. maritimum* (*p* = 6.9 × 10^−1^) or *T. dicentrarchi* (*p* = 1) were present. There was no difference in the number of mortalities during netpen cleaning at LI (*p* = 7.0 × 10^−1^) or when florfenicol was applied (*p* = 5.8 × 10^−2^). 

Comparing each environmental parameter independently to the application of florfenicol at MS, there was a significant relationship with dissolved oxygen at 0 m (Z-value = −4.0, *p* = 5.1 × 10^−5^) and 5 m (Z-value = −3.9, *p* = 1.0 × 10^−4^), as well as the temperature at 0 m (Z-value = 3.8, *p* = 1.4 × 10^−4^), 5 m (Z-value = 4.7, *p* = 2.2 × 10^−6^) and 10 m (Z-value = 4.8, *p* = 2.0 × 10^−6^) using binomial logistic regressions. When the environmental parameters were all considered in an additive format, a relation was observed comparing the water temperature at 0 m (Z-value = 2.5, *p* = 1.2 × 10^−2^) and 5 m (Z-value = 2.0, *p* = 4.2 × 10^−2^). The same comparison for the LI site indicated that when environmental parameters were compared independently, the only variable that was close to generating a significant relationship was the temperature at 0 m (Z-value = −1.8, *p* = 7.7 × 10^−2^). In an additive format, the salinity at 0 m (Z-value = 2.0, *p* = 4.4 × 10^−2^), 5 m (Z-score = 2.0, *p* = 4.7 × 10^−2^) and 10 m (Z-value = 2.4, *p* = 1.8 × 10^−2^) and the interaction of all the variables (Z-value = −2.0, *p* = 4.6 × 10^−2^) also generated a significant relationship. 

Comparing the number of dead fish to environmental parameters at the MS site independently, indicated that dissolved oxygen at 0 m (*p* = 1.6 × 10^−10^), and 5 m (*p* = 4.0 × 10^−10^) and the temperature at 0 m (*p* = 1.2 × 10^−3^), 5 m (*p* = 2.3 × 10^−6^) and 10 m (*p* = 5.1 × 10^−7^) were significantly related; however, all R^2^ values were below 0.25. At the LI site, salinity at 10 m (*p* = 4.2 × 10^−2^), when compared independently, generated a significant relationship; however, the R^2^ was below 0.1. No relationships were observed at either site when all the environmental parameters were included in an additive format.

## 3. Discussion

### 3.1. Isolates

No isolates were identified as *T. maritimum* using qPCR, even though several samples were positive using the MAR assay. *T. maritimum* might exhibit a viable but non-culturable (VBNC) state; increases in temperature and micronutrients such as iron have been reported to have roles in resuscitation of a *Tenacibaculum* isolate [33]. A lack of *T. maritimum* isolates may also be related to a lack of understanding of the exact requirements for selective growth; as media used for culturing *T. maritimum* [2,34] contain complex, undefined chemicals such as seawater [35] or yeast extract [36,37]. *T. maritimum* isolates may also be poorly competitive during isolation and other fast-growing bacteria may inhibit growth, as reported in *Flavobacterium* sp. [38]. In addition, the VBNC state in *Flavobacterium* sp. can be induced by increases in temperature [39] and the absence of nutrients [40]. More research is needed to develop a defined media for isolating specific *Tenacibaculum* species.

All sequenced isolates of *T. dicentrarchi* and *T. finnmarkense* had genetic similarities to isolates linked to finfish mortality events in Chile (*T. dicentrarchi* QCR29 [13], *T. dicentrarchi* TdChD06 [41] and *T. finnmarkense* AY7486TD [27,28,30]). The identification of isolates in BC waters, similar to those found in Chile, indicates that the strains and species are likely found along the West coast of the Americas. It is unknown if *T. dicentrarchi* and *T. finnmarkense* have always had a broad distribution, or if anthropogenic activities, climate change, animal migrations, and oceanic currents have allowed the range of bacteria to expand. Understanding how *Tenacibaculum* disperse through larger geographic scales may allow researchers to understand how netpen sites are colonized by new strains. Research also needs to be placed on developing more specific assays, as the DICEN qPCR assay is also specific to *T. finnmarkense* AY7486TD [31,32].

### 3.2. Seasonal Comparison

Seasonality likely has an important impact on the occurrence of tenacibaculosis, as greater number of *Tenacibaculum* sp., increased fish mortality and more outbreaks were recorded during the spring/summer (MS) compared to the fall/winter (LI). In BC waters, the prevalence of *T. maritimum* in sea lice was the greatest in the summer, which correlated with increases in temperature; however, *T. maritimum* has also been found during the winter [14]. The current study suggests that mortalities and antimicrobial applications to treat mouthrot in the spring/summer were indirectly correlated to decreases in dissolved oxygen and increases in temperature. In contrast, in the fall/winter, decreases in temperature and increases in salinity were correlated to the application of antimicrobials. Both comparisons had a low correlation coefficient; however, similar to a previously noted correlation between increased water temperatures and the abundance of *T. maritimum* [42]. Other variables such as nutrient availability and plankton blooms may also influence the occurrence of tenacibaculosis, as both have correlations to seasonality [43,44] and mortality events [43,44]. Furthermore, understanding the roles of seasonality and dissecting the variables associated with seasonality will allow a greater understanding of *Tenacibaculum* sp. ecology and have downstream implications on managing tenacibaculosis. 

### 3.3. Netpen Cleaning Comparison

Increases in the number of mortalities during netpen cleaning occurred in the spring/summer. Current netpen cleaning practices may be related to tenacibaculosis outbreaks causing direct damage to tissues and facilitating invasion [45]. For example, hydrozoan cnidarians are commonly attached to netpen sites in BC and cleaning practices would expose fish in netpens to the hydrozoan and nematocysts. The stinging nematocysts from *D. typicum* damaged fish tissues and were proposed to facilitate tenacibaculosis [19]. However, regular netpen cleaning is required as excessive biofouling can negatively influence fish health. Research on the implications of modern netpen cleaning practices on salmonid health needs to occur to confirm if the two are indeed related. 

### 3.4. Outbreak Status Comparison

Investigating the distribution of bacteria on numerous samples before, during and after an outbreak is complicated. The lack of positive qPCR samples is one of the main limitations of this study. For future studies, increased sample sizes and a minimum number of positive samples would be valuable. Another limitation is that the pre-, during and post-outbreak status are based on mortality numbers and florfenicol treatment, and the actual outbreak window has likely not been identified.

Before fish were introduced on the site, *T. maritimum* and *T. dicentrarchi* were typically not identified in the environment, with one exception; *T. maritimum* was found at the Larson netpen site. *T. maritimum* and *T. dicentrarchi* were also not identified in newly introduced Atlantic salmon post-smolts; however, a week later at least one of these bacterial species was identified in fish and environmental samples.

Bacteria identified during an outbreak were typically found in greatest abundances on external tissues such as the skin, jaws and gills, while fewer were recorded from internal tissues. It has been proposed that *Tenacibaculum* sp. may make up a portion of the mucosal microbiome of Atlantic salmon in seawater [46], where the presence before and after an outbreak supports that the bacteria should be considered opportunistic pathogens [47,48], potentially inducing tenacibaculosis through dysbiosis [49,50,51]. Of the environmental samples, OO had the most *T. dicentrarchi*, supporting the notion that other organisms associated with netpens can concentrate the bacteria relative to the environment [52,53,54,55]. Given that *Tenacibaculum* can be concentrated in or on OO, they may be useful indicator species for tracking the bacteria and potential tenacibaculosis outbreaks, where indicator species have been widely applied to monitor changes in the environment [56]. More *Tenacibaculum* bacteria or trends for increased numbers were detected from most sample types at MS during an outbreak, except for water samples. Florfenicol in un-consumed pellets and feces could influence the flora of the surrounding environment; as seen in other studies where the microbial diversity decreases [57], or selection for resistant bacteria can occur [58]. When all LI samples were pooled together, an increase in *T. maritimum* was recorded during an outbreak. It should be noted, that *T. maritimum* was not identified in live fish and that more *T. dicentrarchi* was identified at this site during an outbreak. 

After the outbreaks at MS and LI, bacterial numbers decreased or were not detected, except no significant decreases occurred for infrastructure swabs, live fish samples and dead fish samples. Both results support the likelihood for a recrudescence of tenacibaculosis, as has been noted clinically. 

### 3.5. The Primary Agent Responsible for the Larson and Midsummer Outbreaks

From the present study, there is more evidence that *T. dicentrarchi* (*T. finnmarkense*) was the cause of tenacibaculosis at netpen sites. This is supported by the findings that similar numbers of *T. dicentrarchi* were recorded between sites, more *T. dicentrarchi* were recorded throughout an outbreak at both sites in comparison to *T. maritimum*, more *T. dicentrarchi* were recorded during an outbreak and more *T. dicentrarchi* were detected from dead fish compared to live fish. The increase of *T. dicentrarchi* bacteria in dead fish (10-fold increase compared to live fish), is similar to research involving *Flavobacterium* species [59,60]. With numerous studies supporting the role of *T. maritimum* as a fish pathogen, it was likely involved in the outbreaks observed but possibly not as the most important agent. Describing the microbiomes of fish important to aquaculture is necessary, especially if numerous *Tenacibaculum* species can co-occur in the same microbiome and if shifts in populations may be related to infections, as seen with *Vibrio* sp. infections in *C. gigas* [61]. 

## 4. Materials and Methods 

The methods described below are from the master′s thesis by Nowlan (2020) [31].

### 4.1. Sites Used in the Study

Two commercial netpen sites (Midsummer [MS] & Larson Island [LI]) located in the Broughton Archipelago (BC) were selected for sampling. The study was designed to follow groups of post-smolt Atlantic salmon (*Salmo salar*) from introduction to saltwater up to ~500 g. 

### 4.2. Sample Collection, Processing, and Preservation

Over the five-month period from April 2019–September 2019, 14 sets of samples were collected from the MS netpen site. Over the four-month period from November 2019–February 2020, eight sets of samples were collected from the LI nepten site. Samples were collected before, during and after mouthrot outbreaks. All samples were collected in biological triplicates and samples were grouped within three tiers: Tier 1 (environment and fish); Tier 2 (water, infrastructure swabs, organisms associated with netpens (OO), and euthanized and deceased fish); Tier 3 (water-0 m, water-5 m, water-10 m, wood, metal, concrete, plastic, netpen, rubber, invertebrates, primary producers [macroalgae or marine angiosperms, jaws, skin, gill, muscle, head-kidney and spleen). 

At the MS site, samples consisted of infrastructure swabs (netpen surfaces, plastic, metal, wood and concrete) using sterile cotton swabs, 1 L water samples within netpens (0, 5 and 10 m) using a Van-Dorn bottle, OO (primary producers [macroalgae and marine angiosperms] and invertebrates) and fish samples (euthanized and deceased [skin, upper jaw, lower jaw, muscle, spleen, and head-kidney]). Samples collected from the LI site were similar with several exceptions: rubber infrastructures were swabbed instead of wood; primary producers were not collected; only one invertebrate species (likely *Obelia geniculata*) was collected; and for fish tissues, the skin, jaws, gills, spleen and head-kidney were collected. Collected swabs and tissues were stored in RNAlater™ (SIGMA, R0901-500ML), and water samples were placed in Nalgene bottles. Back at Vancouver Island University, water samples were filtered through 180, 20, and 0.22 µm filters simultaneously and the 0.22 µm filters were placed in RNAlater™. All samples stored in RNAlater™ were frozen at −80 °C until needed. Information on specific collected samples can be found in ‘Supplementary material E and F’ by [31].

### 4.3. Bacterial Isolate Collection 

Sterile inoculating loops were used to swab fish tissues and infrastructure; loops were then streaked onto *Flexibacter maritimus* medium (FMM) supplemented with kanamycin to a final concentration of 50 μg mL^−1^. Ten microlitres of pre-filtered water from each depth (0, 5, 10 m) was aliquoted onto FMM agar with kanamycin and was distributed using a curved sterile glass rod. Pure colonies thought to be *Tenacibaculum* underwent Gram staining and DNA extractions. Collected isolates are found in ‘Supplementary material G’ supplied by [31]. Isolates were stored in FMM with 25% glycerol and frozen at −80 °C. Several isolates, based on qPCR results, were sent for sequencing (Molecular Biology Facility, University of Alberta) using generic 16S rDNA primers (27F, 1492R; [62]).

### 4.4. DNA Extractions and Normalization

DNA was extracted from all samples using the OMEGA E.Z.N.A Tissue DNA extraction kit (Omega Bio-tek, Inc., USA) and deviations from the manufacturer’s guidelines are described by Nowlan (2020) [31]. Extracted DNA from most samples was diluted to a concentration of ~14.5 ng µL^−1^ if possible. 

### 4.5. qPCR Application

All the samples underwent *T. dicentrarchi* (DICEN [31,32]) and *T. maritimum* (MAR [18,31]) specific qPCR assays, copying the thermal profile, reagents and equipment used for both assays. For reactions in each well using the DICEN and MAR assay, there was: 10 µL of Probes Master (Catalogue number: 04887301001 [Roche, Switzerland]); 1 µL of 10 µM forward primer; 1 µL of 10 µM reverse primer; 1 µL of 1 µM probe; and 7 µL (~100 ng) of template DNA. All samples were run in triplicate and plates used for qPCR had a positive control, a no template control. 

### 4.6. Conversion of Cq to the Number of Bacteria Per Sample 

Several standard curves described below were used to calculate the theoretical number of bacteria per amount of DNA (100 ng) added to each well and can be found in [31,32]. The standard curve used in this study for fish tissues used an average of both the spiked muscle and head-kidney tissue standard curves from [31,32].

For the DICEN assay, the number of bacteria in each well from environmental samples was calculated using Equation (1):x = (10^ ((Mean Cq of the sample −47.62)/−3.68))/10(1)

For the DICEN assay, the number of bacteria in each well from fish samples was calculated using Equation (2):x = (10^ ((Mean Cq of the sample −44.55)/−3.19))/10(2)

For the MAR assay, the number of bacteria in each well from environmental samples was calculated using Equation (3):x = (10^ ((Mean Cq of the sample −36.68)/−3.45))/10(3)

For the MAR assay, the number of bacteria in each well from fish samples was calculated using Equation (4):x = (10^ ((Mean Cq of the sample −35.38)/−3.42))/10(4)

From the resultant calculations (Equations (1)–(4)), ‘x’ is the number of bacteria per well.

From Equations (1)–(4), ‘x’ was converted to the number of bacteria per sample (x1) using Equation (5):x1 = (x/y * concentration of the DNA extracted from the original sample [ng μL^−1^] * 100 [μL])(5)

Using Equation (5), ‘y’ is the amount of DNA loaded into the well. For most samples, ‘y’ is 100 [amount of DNA added to each well]. 

A final calculation based on the type of sample occurred to achieve standardized units. For swabs, ‘x1′ was not transformed because the whole swab was used in the DNA extraction and the units became the number of bacteria per swab. For water samples, the result from ‘x1′ was multiplied by five, as only a fifth of the filter was used for DNA extractions, and the units became the number of bacteria per litre. For fish tissues and tissues from other organisms associated with netpens, ‘x1′ was multiplied by 33.3 (30 mg of tissue were used) to provide the number of bacteria per g of tissue. All qPCR Cq values and conversions can be found in “Supplementary material E and F” supplied by [31].

### 4.7. Supplementary Data 

Additional data sets obtained during the sampling periods were provided by MCW personnel. These included mortality data (number per site per day), florfenicol treatment (dates and feeding rate), netpen cleaning (date and pen) and various environmental parameters. Environmental data included temperatures (°C) at 0, 5 and 10 m, salinity (‰) at 0, 5, and 10 m and dissolved oxygen (mg L^−1^) at 0 and 5 m.

### 4.8. Statistical Analysis 

Welch′s ANOVA and Games-Howell post hoc test were applied comparing the log-number of bacteria (LNOB) to a single factor (site, assay, outbreak category, sample, sample type, or sample specifics). 

A Kruskal–Wallis test was used to determine if there are differences in the average number of mortalities when florfenicol treatment occurred, when qPCR identified either target bacterial species and when netpen cleaning was applied. A non-parametric generalized additive model (GAM) was used to determine if there was a relationship between the number of mortalities and environmental parameters. Finally, binomial logistic regressions were used to interpret if there was a relationship between the application of florfenicol and any environmental parameter. For all comparisons, a *p*-value ≤ of 0.05 was selected to denote significant differences.

## Figures and Tables

**Figure 1 pathogens-10-00414-f001:**
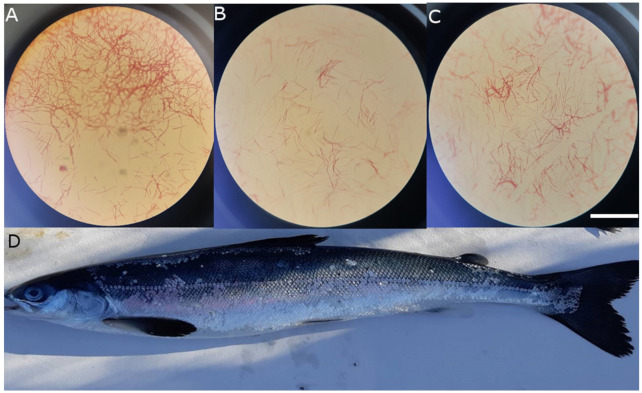
Gram stains of *Tenacibaculum finnmarkense* AY7486TD isolates (LI C6 FM3-M [A, LI C6 FM3-G [B, and LI C6 FM3-F [C]) collected from an Atlantic salmon (**D**, fork length = 32 cm) at the Larson Island netpen site on 1/28/2020 during a tenacibaculosis outbreak. Isolates were identified based on morphology, qPCR results and the 16S rDNA sequence of LI C6 FM3-F. Numerous dislodged scales on the body, areas of discolouration and yellow plaques on the jaws were present. Gram-negative and filamentous bacteria were grown on FMM agar with kanamycin and viewed at ×1000 magnification. The white scale bar (32 µm) applies to (**A**–**C**).

**Figure 2 pathogens-10-00414-f002:**
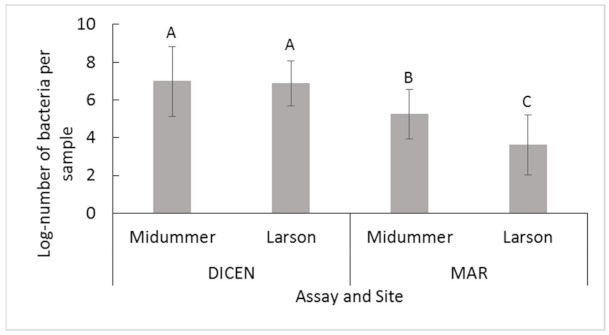
Mean +/− SD. Log-number of bacteria for positive samples for the *T. dicentrarchi* (DICEN) and *T. martimum* (MAR) specific qPCR assays at the Midsummer and Larson Island netpen sites. Different letters indicate significant differences (*p* ≤ 0.05).

**Figure 3 pathogens-10-00414-f003:**
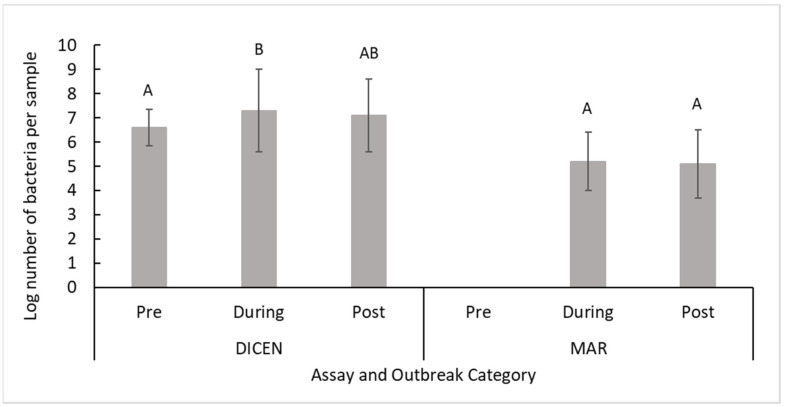
Mean +/− SD. Log-number of bacteria for positive samples from the Midsummer site based on outbreak category using the *T.dicentrarchi* (DICEN) or *T. maritimum* (MAR) specific qPCR assays. Significant differences (*p* ≤ 0.05) are identified using different letters, uppercase and lowercase letters indicate separate Welch′s ANOVAs.

**Figure 4 pathogens-10-00414-f004:**
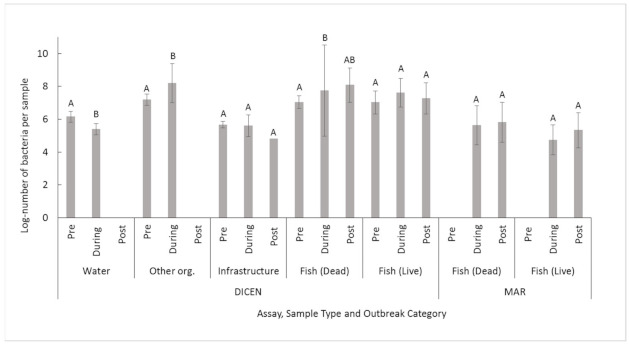
Mean +/− SD. Log-number of bacteria for positive samples from the Midsummer site using the *T. dicentrarchi* (DICEN) or *T. maritimum* (MAR) specific qPCR assay compared against outbreak category for each sample type. Significant differences (*p* ≤ 0.05) are identified using different letters and coloured columns, separate Welch′s ANOVAs occurred for individual sample types.

**Table 1 pathogens-10-00414-t001:** Most similar BLAST ***** comparisons for 16S rDNA sequence of *Tenacibaculum* isolates cultured from Atlantic salmon (*Salmo salar*) including alignment length (bp), query cover, E-value and percent identity.

Isolate Name	Most Similar BLAST Match	Alignment Length (bp)	Query Cover (%)	E-Value	Percent Identity (%)
MS C7 M2	*T. dicentrarchi* QCR29	1326	100	0	99.4
MS C9 F1	*T. dicentrarchi* TdChD06	1335	100	0	99.63
MS C10 M2	*T. finnmarkense* AY7486TD	1335	100	0	100
LI C6 FM1-G	*T. finnmarkense* AY7486TD	1018	100	0	100
LI C6 FM1-F	*T. finnmarkense* AY7486TD	1330	100	0	99.92
LI C6 FM2-G	*T. finnmarkense* AY7486TD	1335	100	0	100
LI C6 FM3-F	*T. finnmarkense* AY7486TD	1335	100	0	99.85

***** Basic Local Alignment Search Tool (BLAST; https://blast.ncbi.nlm.nih.gov/Blast.cgi (accessed on 21 February 2020)).

**Table 2 pathogens-10-00414-t002:** Number of positive qPCR samples at the Midsummer and Larson Island netpen sites using the *T. dicentrarchi*, *T. maritimum* or both specific quantitative-PCR assays. Numbers in brackets represent percentages. Percentages are calculated by dividing the number of positive qPCR samples against the number of samples in total, for fish tissues or for the environment (Env.).

Site	*T. dicentrarchi*	*T. maritimum*	Both
Midsummer	Total: 131 (16.9)	Total: 82 (10.6)	Total: 31 (3.99)
	Fish: 71 (17.9)	Fish: 76 (19.2)	Fish: 30 (7.59)
	Env.: 60 (15.7)	Env.: 6 (1.57)	Env.: 1 (0.262)
Larson Island	Total: 34 (8.79)	Total: 24 (6.20)	Total: 6 (1.55)
	Fish: 23 (13.8)	Fish: 9 (5.39)	Fish: 6 (3.59)
	Env.: 11 (5.09)	Env.: 15 (6.94)	Env.: 0

**Table 3 pathogens-10-00414-t003:** Statistical comparisons (Welch′s ANOVA) using the Midsummer (MS) and Larson Island (LI) netpen sites and using the *T. dicentrarchi* (DICEN), *T. maritimum* (MAR) or both specific qPCR assays. For each level of a comparison there is the mean log-number of bacteria (LNOB) and SD *, for each comparison there is also an attributed F-value and corresponding *p*-value (*p*).

LNOB Comparison	Mean LNOB ± SD *	F Value	*p*
1. Between sites using both assays	MS: 6.3 ± 1.7 ^A^ LI: 5.6 ± 2.0 ^B^	F_1,95.18_ = 7.7	0.0068
2. Within/between sites comparing DICEN and MAR assays	MS DICEN: 7.1 ± 1.5 ^A^ MS MAR: 5.2 ± 1.3 ^B^ LI DICEN: 6.8 ± 1.2 ^A^ LI MAR: 3.9 ± 1.7 ^C^	F_3,91.88_ = 63	<2.2 × 10^−16^
3. Throughout an outbreak using both sites and assays	Pre: 5.7 ± 1.9 ^A^ During: 6.5 ± 1.8 ^B^ Post: 5.4 ± 1.4 ^A^	F_2,110.44_ = 13	1.2 × 10^−5^
4. Between fish and the env. using both sites and assays	Fish: 6.5 ± 1.7 ^A^ Env.: 5.3 ± 1.7 ^B^	F_1,164.11_ = 33	4.9 × 10^−8^
5. Between sample types using both sites and assays	Fish-Live: 6.0 ± 1.5 ^A^ Fish-Dead: 6.8 ± 1.8 ^B^ OO: 7.0 ± 1.7 ^A,B^ Water: 4.5 ± 1.5 ^C^ Infrastructure: 5.0 ± 1.3 ^C^	F_4,80.43_ = 23	7.8 × 10^−13^
6. Throughout an outbreak at MS using the DICEN assay	Pre: 6.6 ± 0.74 ^A^ During: 7.3 ± 1.7 ^B^ Post: 7.1 ± 1.5 ^AB^	F_2,13.481_ = 7.0	0.0082
7. Throughout an outbreak at MS using the MAR assay	Pre: NA During: 5.2 ± 1.2 ^A^ Post: 5.1 ± 1.4 ^A^	F_1,77.501_ = 0.053	0.82
8. Between sample types at MS using the DICEN assay	Fish-Live: 7.3 ± 0.81 ^A^ Fish-Dead: 7.9 ± 1.6 ^A^ OO: 7.8 ± 0.50 ^A^ Water: 5.6 ± 1.5 ^B^ Infrastructure: 5.6 ± 0.49 ^B^	F_4,49.523_ = 60	<2.2 × 10^−16^
9. Between sample types at MS using the MAR assay	Fish-Live: 4.9 ± 1.1 ^A^ Fish-Dead: 5.7 ± 1.2 ^B^ OO: 3.8 ± 0.65 ^A,B,C^ Water: 2.8 ± 0.91 ^C^	F_3,6.73_ = 15	0.0021
10. In live fish throughout an outbreak at MS using the DICEN assay	Pre: 7.1 ± 0.78 ^A^ During: 7.5 ± 0.81 ^A^ Post: 6.7 ± 0.30 ^A^	F_2,4.5048_ = 4.08	0.097
11. In infrastructure swabs throughout an outbreak at MS using the DICEN assay	Pre: 5.6 ± 0.15 ^A^ During: 5.6 ± 0.61 ^A^ Post: 4.8	F_1,25.651_ = 0.29	0.59
12. In dead fish throughout an outbreak at MS using the DICEN assay	Pre: 6.9 + 0.270 ^A^ During: 8.2 + 1.75 ^B^ Post: 8.1 + 1.04 ^A,B^	F_2,5.27_ = 12	0.011
13. In OO throughout an outbreak at MS using the DICEN assay	Pre: 7.1 ± 0.316 ^A^ During: 8.2 ± 1.18 ^B^ Post: NA	F_1,8.76_ = 5.3	0.047
14. In water throughout an outbreak at MS using the DICEN assay	Pre: 6.1 ± 0.316 ^A^ During: 5.3 ± 0.361 ^B^ Post: NA	F_1,11.50_ = 19	0.0011
15. In dead fish throughout an outbreak at MS using the MAR assay	Pre: NA During: 5.7 ± 1.22 ^A^ Post: 5.7 ± 1.14 ^A^	F_1,48.76_ = 0.00077	0.98
16. In live fish throughout an outbreak at MS using the MAR assay	Pre: NA During: 4.9 ± 1.14 ^A^ Post: 4.9 ± 0.837 ^A^	F_1,27.155_ = 0.011	0.91
17. Throughout an outbreak at LI using the DICEN assay	Pre: 9.4 During: 6.8 ± 1.14 Post: 6.1	NA	NA
18. Throughout an outbreak at LI using the MAR assay	Pre: 2.7 ± 0.831 During: 5.0 ± 1.65 Post: 5.3 ± 1.10	F_2,9.6835_ = 17	6.4 × 10^−4^
19. Between sample types at LI using the DICEN assay	Fish-Live: 6.5 ± 0.469 ^A^ Fish-Dead: 7.7 ± 1.21 ^B^ OO: 7.1 ± 0.678 ^A,B^ Water: NA Infrastructure: 5.2 ± 0.469 ^C^	F_3,12.80_ = 20	4.6 × 10^−5^
20. Between sample types at LI using the MAR assay	Fish-Live: NA Fish-Dead: 5.1 ± 1.41 ^A^ OO: NA Water: 3.0 ± 0.71 ^B^ Infrastructure: 2.3 ± 0.77 ^B^	F_2,16.597_ = 19	4.7 × 10^−5^

* Different superscript letters after the SD represent differences between levels using the Games-Howell post hoc test. No SD after a reported mean indicates the value for a single positive sample. NA= Not available, env. = Environment, OO = other organisms associated with netpens.

## Data Availability

Data can be found through contacting the corresponding author.

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
