# Peer review of "Application of Quantitative-PCR to Monitor Netpen Sites in British Columbia (Canada) for Tenacibaculum Species"

_pathogens, 2021, doi:10.3390/pathogens10040414_

Round 1

Reviewer 1 Report

Nowlan et al. have used quantitative-PCR assays to detect T. maritimum and T. dicentrarchi at commercial Atlantic salmon (Salmo salar) netpen sites before, during and after tenacibaculosis outbreaks. This study describes spatial and temporal distributions of Tenacibaculum species at above mentioned sites which may aid in the management of tenacibaculosis. It sounds interesting research and data, and I am sure it will help us to increase our knowledge about this important pathogen.

I have a number of comments as follows,

- Obviously, this paper base on quantification method but unfortunately, I found little information in methods part regarding how quantification has been done in detail. Although authors have a lot of referencing to previous works from same lab, I still think every paper should talk for itself specially if methods and analytical parts are essential for understanding results. Moreover, I found that Nowlan 2020 work is a master thesis and not peer reviewed paper; then it make more important to have method in detail in this work as well! However, I also enjoyed reading the thesis (Nowlan 2020) and applaud the author for using detailed information on every step.

- I suggest fitting more results in abstract regarding bacterial dynamics throughout outbreak!

- I found introduction so short and I believe it would be helpful to mention previous MLSA work on this species and stating main advantage of your method over such detailed previous works!

- Discussion is not concise and does not flow well between findings. Since your research has a number of important implications, discussion should be refined to draw out the key outcomes and discuss those within the context of applied aquaculture; for example, what additional, future work may be warranted? What recommendations can be drawn?

- Line 13-14 “Both bacteria were also found in invertebrates, including cnidarians, molluscs and crustaceans, and these may be useful as indicator species for Tenacibaculum”, please make it clear if this is a known knowledge or part of your results? If so, has any data presented in this manuscript? Also, I could not find any data for such important results stated in line 19-20.

- Move Table 1 to supplementary material!

- Please explain more about numbers in lines 75-76, Again this information should be in methods and in detail: technical and biological replicates and so on?

- Line 259: I donot think microbiome fitted here please reword this for exampling co-occurring species or coinfection.

-  Line 270-272: “Table 1”: but Table 1 has Blast information not sampling details!

- Line 270: why authors have decided to sample warm months of year from one site and colder months in another. Is there possibility of seasonal sampling from one site? If so, results upon seasonality should be touched cautious!

- In all graphs logarithmic number of bacteria has been shown, I believe this would be enough for all comparisons, but would you please make it more clear what would the real number range that you quantified.

- I also could not find data and tables to one of the main results in this manuscript that is devoted to relationships between fish mortalities/antimicrobial use and water quality parameters.

- In results section 2.3 supplemental data comparisons: please make it more clear for readers what do you mean by significant relationships? Add figures for this results or clearly state the direction of relationship (positive ,negative,..).

- Line 305: remove error text!

Reviewer 2 Report

In this work the occurrence of  T. maritimum and T. dicentrarchi was studies before, after and during outbreaks disase. The work is interesting but its interpretation is so complicate since sampling strategy is not showed in the beginning of the results. Sampling strategy (where/when the samples were obtained?) must be clearly shown in the beginning of the result. This information is crucial for the interpretation of the work. In the same way, Materials and method section must be before results. 

Reviewer 3 Report

MDPI

Pathogens-1159

In this manuscript, the authors used quantitative-PCR assay and detected two species, Tenacjbaculum maritimum and T dicebtrarchi at commercial Atlantic salmon netpen sites. Results indicated that significant increases in T dicebtrarchi occurred during mortality event in summer, supporting arole for this species in the pathogenesis of tenacibaculosis.  In this manuscript, the authors have reported that Tenacibaculum species appear to have a ubiquitous spatial and temporal distribution around stocked sites. This is well designed and well written manuscript. The authors have provided novel information in Atlantic salmon farm. Check references 6 and 19 “PLoS ONE” and “”PloS one”. Check references 22 and 38 “Genome Announc” and “Genome Announce”.

Round 2

Reviewer 2 Report

I congratulate the authors for the work done